# Antioxidative Potential of Red Deer Embryos Depends on Reproductive Stage of Hind as a Oocyte Donor

**DOI:** 10.3390/ani10071190

**Published:** 2020-07-14

**Authors:** Anna J. Korzekwa, Angelika M. Kotlarczyk, Agata A. Szczepańska, Martyna Grzyb, Alicja Siergiej, Izabela Wocławek-Potocka

**Affiliations:** 1Department of Biodiversity Protection, Institute of Animal Reproduction and Food Research of Polish Academy of Sciences (IAR&FR PAS), Tuwima 10 Str., 10-748 Olsztyn, Poland; a.kotlarczyk@pan.olsztyn.pl (A.M.K.); a.szczepanska@pan.olsztyn.pl (A.A.S.); m.grzyb@pan.olsztyn.pl (M.G.); 2Department of Gamete and Embryo Biology, IAR&FR PAS, Tuwima 10 Str., 10-748 Olsztyn, Poland; a.siergiej@pan.olsztyn.pl (A.S.); i.woclawek-potocka@pan.olsztyn.pl (I.W.-P.)

**Keywords:** oxidative potential, blastocyst, embryo, red deer

## Abstract

**Simple Summary:**

Deer breeding tends to select animals for obtain the high meat quality and in case of males preferred shape and weight of antlers. Fertilization in vitro (IVF) using high-indexing parents results favorable features. Moreover, evaluation of effective method of IVF on *Cervus elaphus* as a model, will be useful for application on Cervids in danger of extinction. The effectivity of IVF depends on quality of gametes and proper development of embryo. The aim was to compare the blastocyst stages of red deer embryos in respect of IVF efficiency, morphology, apoptotic and proliferative abilities, and antioxidative potential according to the reproductive status of hinds. We used three experimental groups, including the ovaries collected post mortem on the 4th and 13th days of the estrous cycle (farmed animals) and during pregnancy (wild animals). Frozen-thawed epididymal semen was used for IVF. Blastocyst quality, apoptotic, and antioxidative potential of blastocysts were evaluated. Results indicate that red deer embryos on blastocyst stage received in vitro collected from hinds on 4th day of the estrous cycle as an oocyte donor are characterized by better antioxidative potential and qualities to those developed from oocytes collected from hinds on 13th day of the estrous cycle and pregnancy.

**Abstract:**

The aim was to compare the blastocyst stages of red deer embryos in respect of in vitro fertilization (IVF) efficiency, morphology, apoptotic and proliferative abilities, and antioxidative potential according to the reproductive status of hinds. We used three experimental groups, including the ovaries collected post mortem on the 4th and 13th days of the estrous cycle and during pregnancy (*n* = 18). After oocyte maturation, frozen-thawed epididymal semen was used for IVF. Blastocyst quality, apoptotic potential by determining the mRNA expression of *BAX*, *BCL-2*, *OCT4*, *SOX2*, and placenta-specific 8 gene (*PLAC8*)*,* and antioxidative potential of blastocysts were evaluated by determining the mRNA expression of *CuSOD*, *MnSOD*, and *GPX* as well as the enzymatic activity of superoxide dismutase and reduced glutathione. The highest development rate of expanded blastocyst, mRNA expression of *BCL-2*, *OCT4*, *SOX2*, and *PLAC8* and mRNA expression and enzymatic activity of the antioxidative factors increased (*p* < 0.05) in blastocysts developed from the oocytes collected on the 4th day, compared to those developed from the oocytes collected on the 13th day of the cycle and during pregnancy. Our study indicates that the 4th day of the estrous cycle is the most effective period for oocyte collection for IVF and embryo development in hinds, considering quality parameters and antioxidative potential of the blastocysts.

## 1. Introduction

Red deer (*Cervus elaphus* L.) is currently one of the most widespread European ungulate species [1]. Nowadays, the New Zealand is the kingdom of deer farms [2], but they have also been developed in Europe during the last twenty years. Deer breeding is focused on animal selection to obtain the preferred meat quality. Furthermore, males are preferred for trophy hunting and their ability to prolong the reproductive season. According to Asher and Pearse 2002, the improvement of the farmed deer reproductive performance is associated with the increased pregnancy rates of the hinds entering puberty and an improvement in their lactational performance of hinds [3]. Thus, the selection of high-indexing parents results in the offspring with favorable features and high in vitro fertilization (IVF) efficiency. Further efforts to perform embryo transfer and production from this selected deer are oriented toward the acceleration of their genetic improvement rates. During the last two years, some papers revealed methodology concerning red deer in vitro production [4,5,6].

Red deer are highly seasonal ruminants with polyestrous females exhibiting the onset of ovulatory activity in autumn [7,8]. Seasonal breeding in ruminants is a survival mechanism that ensures the birth of offspring under the most favorable climatic and food availability conditions [9]. After the last luteal phase of the breeding season in ewe, the decline in serum progesterone (P4) concentrations is not accompanied by a prominent increase in 17-beta estradiol (E2) secretion [10]. The lack of a normal pre-ovulatory peak of E2 might be partly due to the reduction in ovarian responsiveness to gonadotropic hormones [10], but it is primarily caused by an enhanced negative feedback suppression of luteinizing hormone (LH) secretion, mediated by E2 [9]. It is remarkable that despite decreased LH secretion in ewes during anestrous, the growth of large antral follicles continues in a pattern similar to the breeding season [11]. Such knowledge is not available in Cervidae; furthermore, it is not known how the follicles from anestrous phase influence oocyte quality and IVF efficiency. Considering the further potential of the IVF embryos for transfer in red deer, it is targeted to evaluate the optimal period (estrous cycle versus pregnancy) of oocyte collection, which influences the rate of fertilization success in hinds [12]. The most common evaluation of the quality and embryo development potential of oocytes in mammals is based on their morphology [13] and concentration of peripheral blood steroids P4 and E2, which facilitates the assigning of follicles to different groups (atretic, healthy, and transitional) [14]. The effect of reproductive phase, luteal versus follicular, was noted with luteal stage oocytes yielding a higher rate of blastocyst formation. Porcine blastocysts derived from luteal phase oocytes also had higher hatching ability than those derived from follicular phase oocytes [15]. The number and quality of oocytes useful for further maturation and fertilization might differ depending on the reproductive status of hinds. Thus, we compare, in this study, the number and quality of oocytes, IVF efficiency and the blastocyst quality in hinds which were oocyte donors both from luteal phase (the 13th day of the estrous cycle with relatively high P4 blood concentration) and follicular phase (the 4th day of the estrous cycle with relatively high E2 blood concentration) during the breading season and pregnancy. 

Therefore, it is important to determine the average number of ovarian follicles with dense and cohesive cumulus and uniformly granulated cytoplasm as an important criterion of oocyte quality and reproductive competence in hinds. Transcription factors, such as placenta-specific 8 (PLAC8), OCT4, and SOX2, which have been proposed for determining embryo quality parameters in IVF, are expressed in the bovine, ovine, and deer blastocysts obtained from the oocytes used for IVF [6,14,16]. The presence of programmed cell death might lead to oocyte degeneration [17], and early embryonic death [16]. Thus, BCL-2 and BAX expression is commonly used as an apoptosis marker in oocytes and blastocysts in a wide spectrum of species [18,19]. 

Most ovarian studies have focused on oxidative stress during ovulation, luteogenesis, luteal function, and luteolysis in animals [20,21]. Moreover, the follicular analyses of pro- and anti-oxidants have predominantly focused on human follicular fluid samples [22]. Oxidative stress refers to an imbalance between the concentrations of pro- and anti-oxidants, which, in turn, results in compromised cellular functions. Pro-oxidants have extensive sources and can be generated by the products of cellular respiration and steroid metabolism. The oxidative stress during embryo culture has been associated with lower developmental potential [23], changes in gene expression [24], histone remodeling [25], and increased apoptotic rates [26]. Oxidative stress is a cellular condition that may play a role in early embryo development, especially when oocytes are collected from females in different reproductive phases (estrous/anestrous/pregnancy), however, currently, there is a lack of direct supporting evidence. Several antioxidant enzymes are involved in the protection of embryos against oxidative stress. Superoxide dismutases (SODs) allow superoxide radicals to be scavenged and are involved in the first enzymatic pathway that protects cells against toxic oxygen radicals and their by-products. Subsequently, hydrogen peroxide is eliminated by either catalase or glutathione peroxidase (GPX), which reduce lipid hydroperoxides [27]. SOD exists in three isoforms with varying structures, regulatory mechanisms, localizations, and functions. The Cu-SOD and Zn-SOD (SOD1) are typically cytosolic, and the Mn-SOD (SOD2) is mitochondrial, while the extracellular SOD (SOD3) scavenges superoxide radicals in extracellular fluids and spaces [28]. According to Sanchez–Ajofrin et al. oxygen tension during in vitro oocyte maturation and fertilization affects embryo quality in red deer [6]. 

Therefore, the present study was designed to test the following hypotheses: (1) the red deer oocytes collected for IVF exhibit different developmental competencies measured by IVF effectivity, competence and development rates of blastocyst, and qualities evaluated by determination of mRNA expression of crucial blastocyst development markers depending on their reproductive status and (2) the blastocysts collected from oocytes during estrous cycle and pregnancy present different potencies against oxidative stress. To test these hypotheses in the estrous cycle and pregnancy of hinds, we evaluated the blastocyst rates and compared the number of blastocysts, considering the cumulus-oocyte complexes (COCs) collected from one ovary and the time of the first cleavage. Furthermore, we determined the mRNA levels of the factors influencing oocyte quality (PLAC8, OCT4, and SOX2) and apoptosis (BAX and BCL-2) and the activity of enzymes SOD-1, SOD-2, and GPX of in vitro cultured embryos in the blastocyst stage. 

## 2. Materials and Methods

### 2.1. Oocyte Collection

Twelve 4-year–old red deer females were used for the post mortem collection of ovaries directly after slaughter at a deer farm in Rudzie near Gołdap (North-East Poland). The ovaries were collected from hinds during the estrus phase, i.e., on the 4th (*n* = 6; the first experimental group) and 13th (*n* = 6; second experimental group) days of the estrous cycle after pharmacological synchronization of the animals. The induction of estrus and ovulation in hinds during the estrous cycle was performed by applying a double controlled internal drug-release (CIDR) insert (*Pfizer* Animal Health, New York, NY, USA; *1.38* g of P4), using a 12–day regimen of intravaginal CIDR devices. For better synchronization, the device was replaced after 7 days to maintain the luteal concentration of P4 until the end of the treatment period. Additionally, 200 IU of human chorionic gonadotropin (hCG; Folligon, Intervet, International B.V., Boxmeer, Holland) was injected intramuscularly on day 12 [29]. The estrus was observed 54–56 h after the second CIDR insert removal in the hinds. The day of the estrous cycle was evaluated by macroscopically observing the ovaries and uterus and confirmed by determining E2 and P4 levels in the blood plasma using radioimmunoassay (RIA). The blood samples were collected from the heart. The collection of experimental material was performed from these two experimental groups on the same day, and the ovaries and the blood were brought to the laboratory simultaneously. The reasons for culling animals from the herd in the farm were; economic considerations and herd renewal. All veterinary procedures were conducted after receiving the agreement of Local Ethical Committee in Olsztyn (Poland, Agreement Number 7/2019).

The ovaries were also collected from wild pregnant hinds (*n* = 6; third experimental group) without any pharmacological synchronization during the hunting season (November 3–6, 2019). The ovaries were collected from 4-year-old hinds at ca. 40 days of gestation (Forest District Strzałowo, N-E Poland) based on embryo morphology and development and climatic conditions connected with local air temperature decrease during the rut noticed by experienced hunters, 10–15 min after the shot (license number: ZG–7521–3/2019). The presence of an embryo in the uterus confirmed the pregnancy in hinds. The age of hinds was assessed on the basis of a tooth clash by a specialist—a hunter participating in the hunt—who had appropriate authorization in this field (Main Warmia and Mazuria District hunter), and, further, the age was confirmed after laboratory analyses of elemental mineral composition of the basic tooth-building elements. The peripheral blood samples were collected from the heart for E2 and P4 determinations by RIA. Based on the seasonality of red deer, experimental material collection from pregnant hinds was not possible at the same time as from the another group of hinds, and procedure of further experiment conducting to the fertilization was provided separately for this third experimental group. 

The ovaries were transported to the laboratory in sterile phosphate buffered saline (PBS) in 20 °C. Time between collection and processing was about 70 min. 

The cumulus-oocyte complexes were obtained by aspiration from the subordinate ovarian follicles, less than 5 mm in diameter, (the first experimental group) and by maceration of the ovarian tissue from the same ovary after aspiration (the second experimental group) from the hinds on the 4th and 13th days of the estrous cycle and from pregnant females. Thereafter, a stereo microscope (Discovery V20, Zeiss, Poznan, Poland; SZX7, Olympus, Warsaw, Poland), was used to identify the COCs consisting of oocytes with homogeneous ooplasm without dark spots and surrounded by at least three layers of compact cumulus cells for the study. The COCs were washed twice in the wash medium (61008; IVF Bioscience, Falmouth, UK) and were subsequently washed in maturation medium (61002; IVF Bioscience, Falmouth, UK). The procedures of in vitro oocyte maturation (IVM) and IVF were conducted according to Boruszewska et al. [30]. 

### 2.2. In vitro Oocyte Maturation

Groups of 20 immature COCs in each from three experimental group were placed into “Perti” dishes containing 100 µL of maturation medium overlaid with mineral oil (M8410, Sigma, Poznań, Poland) and incubated at 38.5 °C in a 5% CO_2_ humidified air atmosphere for 23 h. Thereafter, the COCs were washed in fertilization medium (61003; IVF Bioscience, Falmouth, UK).

### 2.3. Semen Collection, Cryopreservation, and Preparation for in vitro Fertilization

The epididymal semen was obtained post mortem from wild seven bulls shot during the rut (September/October 2019) in Strzałowo Forestry (7 years old, based on antler shape and clash of teeth, evaluation was provided by the same hunters as for wild hinds, individuals non-dominant in the group of bulls, North–East, Poland). For IVF, the frozen-thawed semen from the same bull was used throughout the experiment. The semen was diluted with the commercial extender Bioxcell (IMV). Immediately after collection, the white ejaculate fraction was diluted 1:3. Then the concentrations and mobility of spermatozoa were determined. The average concentration of non-diluted spermatozoa in the ejaculates was 6.93 × 10^9^ ± 0.73/mL, and the motility was 95% (CASA; Hamilton-Thorne Biosciences, Beverly, MA, USA). The spermatozoa were cryopreserved according to Demianowicz et al. [31]. In details, sperms equilibrated at 4 °C during 4 h, and, after, loaded into straws (0.25 mL). Process of freezing was carried out for 10 min on polystyrene frames 4 cm above the surface of liquid nitrogen using Minicool 40PC (Air Liquide). The programmed the freezing rate was, respectively: In the range of + 4 °C to −10 °C 5 °C/min., −10 °C to −100 °C 40 °C/min., and −100 °C to −140 °C 20 °C/min. Semen post-thaw average motility was 71%. The semen from the bull with the best sperm motility (77%) was used for IVF. The percentage of morphologically abnormal sperm used for IVF was below 5%. 

### 2.4. In vitro Fertilization

Immediately after thawing, the semen was layered under the capacitation medium (61004; IVF Bioscience, Falmouth, UK) to allow for motile sperm recovery using a swim-up procedure. After incubation, the semen was double centrifuged at 200× *g* for 5 min, the supernatant was removed, and the sperm pellet was diluted in an appropriate volume of fertilization medium to a final concentration of 2 × 10^6^ motile sperms/mL. Groups of 20 COCs in each of three experimental groups were co-incubated with spermatozoa in drop in Petri dishes containing 100 µL of fertilization medium overlaid with mineral oil for 15 h at 38.5 °C in a 5% CO_2_ humidified air atmosphere. The day of in vitro insemination was considered as day 0. The embryos were separated from the cumulus cells by vortexing and washing three times in the wash medium. The cleavage rates were assessed after 48 h, and the embryos were cultured in drop in Petri dishes containing 100 µL of culture medium (61001; IVF Bioscience, Falmouth, UK) at 38.5 °C in an atmosphere of 5% CO_2_, 5% O_2_, and 95% N_2_ with high humidity. 

### 2.5. Sample Collection

Among all blastocysts for further experiments only expanded blastocysts in each experimental group (group I—4th day of the estrous cycle, group II—13th day of the estrous cycle, and group III—pregnancy) were collected between the 6th and 9th days. The developmental stage and embryo quality were determined based on the International Embryo Transfer Society (IETS) mannual, with further indicators described by Gardner and Schoolcraft 1999 [30]. The quality of the blastocysts was scored as follows: Grade A, excellent and good; grade B, fair and moderate; grade C, poor; and grade D, dead or degenerating. Embryos classified as quality grade A–C were selected for conducting real-time PCR and for determining enzyme activity. All expanded blastocysts were stored in extraction buffer at −80 °C until further analysis. 

### 2.6. Experimental Design

#### 2.6.1. Preliminary Experiment. Concentration of Progesterone and 17-Beta Estradiol in the Blood Plasma

For the purpose of confirmation of luteal, follicular phase of the cycle, and pregnancy in each experimental group, in the blood plasma, levels of E2 and P4 were determined in the hinds on the 4th and 13th days of the estrous cycle as well as pregnant hinds by RIA.

#### 2.6.2. Experiment 1. Microscopic Evaluation of IVF Effectivity, Blastocyst Quality and Development Depending on the Reproductive Status of Hinds 

The aims are comparison of cleavage rate, total number of collected oocytes used for IVM, total COCs number used for IVF, number of expanded blastocysts collected on particular days post IVF, and development rate of blastocysts and mRNA expression of *BCL-2, BAX*, *PLAC8, OCT4*, and *SOX2* in three experimental groups of hinds-oocyte donors.

##### Microscopic Evaluation of IVF Effectivity and Blastocyst Quality

Cleavage rates were assessed at approximately 36 hpi and blastocyst yield was recorded on 6, 7, 8, and 9 dpi. The evaluation of: Cleavage rate, total number of collected oocytes used for IVM, total COCs number used for IVF, the total number of expanded blastocysts collected between 6th and 9th day of culture, number of expanded blastocysts collected on particular days post IVF, and development rate of blastocysts were proceeded. The grade of blastocysts collected on days 6, 7, 8, and 9 was similar, thus the results are presented as the average from blastocysts collected between 6th and 9th day of culture. Moreover, determination of mRNA expression for examined genes and enzymatic activity in each experimental group was not possible on particular day of blastocyst collection because of the limitation of animal number and experimental material consequently and for this purpose all blastocysts were pulled.

##### Determination of the mRNA Expression of BCL-2, BAX, PLAC8, OCT4, and SOX2

*BAX*, *BCL-2*, *OCT4*, *SOX2*, and *PLAC8* mRNA expression was evaluated in the expanded blastocysts.

#### 2.6.3. Experiment 2. Comparison of the Antioxidative Potential of Red Deer Embryos during the Luteal and Follicular Stage of the Estrous Cycle and Pregnancy

The aim is comparison of the antioxidative potential depending on the reproductive status of hinds—oocyte donors in the expanded blastocysts. A real-time PCR was carried out for determining the mRNA expression of the *CuSOD*, *MnSOD*, and *GPX* genes and a colorimetric assay was used to determine the enzymatic activity of SOD and GSH in expanded blastocysts in three experimental groups: From oocytes collected on the 4th and the 13th day of the estrous cycle and in pregnancy. 

### 2.7. Determinations

#### 2.7.1. RNA Isolation and Reverse Transcription

For total RNA isolation, six pools of five embryos were used for each experimental group. The embryos were suspended in the extraction buffer and were processed for RNA isolation according to the manufacturer's instructions (KIT0204; Arcturus PicoPure RNA Isolation Kit, Applied Biosystems, Thermo Fisher Scientific, Waltham, MA, USA). The DNase treatment was performed for the removal of genomic DNA contamination using the RNase-free DNase Set (79254, Qiagen GmbH, Hilden, Germany). The RNA samples were stored at 80 °C until reverse transcription. The reverse transcription (RT) was carried out using oligo (dT) 12e18 primers (18418–012) and Super Script III reverse transcriptase (18080–044) in a total volume of 20 µL to prime the RT reaction and produce cDNA. The RT reaction was carried out at 65 °C for 5 min and then at 42 °C for 60 min, followed by a denaturation step at 70 °C for 15 min. RNase H (18021–071) was used to degrade the RNA strand of an RNA–DNA hybrid (37 °C for 20 min). The RT products were diluted 10 times and were stored at −20 °C until real-time PCR analysis.

#### 2.7.2. PCR Amplification

A real-time PCR was performed using an ABI Prism 7900 sequence detection system (Applied Biosystems, Foster City, CA, USA) and Maxima^®^ SYBR Green/ROX qPCR Master Mix (K0222, Thermo Fischer Scientific, Waltham, MA, USA). The PCR samples were analyzed in 384-well plates. Each reaction well (10 µL) contained 3 µL of the RT product, 5 mM each of forward and reverse primers, and 5 µL of SYBR Green PCR master mix. In each reaction, we used the cDNA quantity equivalent to 0.25 embryos. A real-time PCR was performed under the following conditions: 95 °C for 10 min, followed by 40 cycles of 94 °C for 15 s and 60 °C for 60 s. Subsequently, melting curves were obtained for each PCR product to ensure single-product amplification. To exclude the possibility of genomic DNA contamination in the RNA samples, the reactions were also performed either with blank-only buffer samples or in the absence of the reverse transcriptase enzyme. The specificity of the PCR products for all tested genes was confirmed by gel electrophoresis and sequencing. The efficiency of the target and internal control amplifications ranged between 95% and 100%. The real-time PCR miner algorithm was used for the relative quantification of the mRNA levels. The mRNA quantification of the examined genes was conducted using the primers specific for *BAX, BCL-2*, *OCT4, SOX2, PLAC8*, *GPX, MnSOD*, and *CuSOD*. The results of mRNA abundance were normalized to the glyceraldehyde-3-phosphate dehydrogenase (*GAPDH*, an internal control) mRNA level and were expressed as arbitrary units. This reference gene was chosen using the Norm-Finder software, comparing three candidate genes: *GAPDH*, *ACTB*, and *RN18S* [32]. The primers were designed using an online software package [33]. The primer sequences and sizes of the amplified fragments for all transcripts are shown in Table 1. 

#### 2.7.3. Antioxidative Activity of Embryos

The activity of antioxidative enzymes SOD-2 (K335, BioVison, Milpitas, CA, USA) and reduced glutathione (GSH; ab239727, Abcam, Cambridge, UK) in embryos was detected using the ELISA kits according to the manufacturer’s instructions. Six pools of five embryos were used for each experimental group. The absorbance values of the resulting reaction mixtures were read at 450 nm using a microplate absorbance reader. The intra/inter-assay coefficients of variation were 2.68%/8.36% for SOD and 2.35%/7.21% for GPX. A solution of 1000 μM hydrogen peroxide in culture medium was used as a positive control for SOD-2 and GPX activity measurement.

#### 2.7.4. Steroid Concentration

The plasma measurements of P4 were performed using RIA (KIP1458, DIAsource ImmunoAssays, Louvain-la-Neuve, Belgium). The standard curve values ranged from 0.12 to 36 ng/mL, and the ED50 of the assay was 0.06 ng/mL. The intra-assay and inter-assay coefficients of variation (CVs) were 6.5% and 8.6%, respectively.

The plasma measurements of E2 were performed using RIA (KIP0629, DIAsource ImmunoAssays, Louvain-la-Neuve, Belgium). The standard curve values ranged from 1 to 355 pg/mL, and the ED50 of the assay was 0.5 pg/mL. The intra-assay and inter- assay CVs were 4.7% and 6.1%, respectively.

All samples were pipetted in duplicate for each run, and validation was performed according to Korzekwa et al. [34]. 

### 2.8. Statistical Analyses

The statistical analysis of the results was performed using GraphPad Prism (GraphPad PRISM, Version 8.3.0, San Diego, CA, USA). The relationship between the mRNA relative abundance and enzyme activity was determined using one-way analysis of variance (ANOVA) followed by a Bonferroni post hoc test. The statistical analysis involved a comparison between blastocysts developed from the oocytes collected on the 4th and 13th days of the estrous cycle and during pregnancy. All numerical data were expressed as the arithmetic mean ± SEM. 

The cleavage and blastocyst rates (%) were analyzed by chi-squared test with Yates' correction. Differences at *p* < 0.05 were considered statistically significant.

## 3. Results

### 3.1. Preliminary Results. Concentration of Progesterone and 17-Beta Estradiol in Blood Plasma

The P4 concentration in the blood plasma was approximately above 190% in hinds on the 4th day of the estrous cycle, compared with the hinds on the 13th day of the estrous cycle and pregnant hinds (*p* < 0.001; Figure 1). 

The concentration of E2 increased above 50% in hinds on the 4th day of the estrous cycle and in pregnant hinds, compared with those on the 13th day of the cycle (*p* < 0.05; Figure 1).

### 3.2. Experiment 1. Microscopic Evaluation of IVF Effectivity, Blastocyst Quality and Development Depending on the Reproductive Status of Hinds 

#### 3.2.1. Microscopic Evaluation of IVF Effectivity and Blastocyst Quality

The most effective period of the reproductive phase for IVF in hinds was found to be the 4th day of the estrous cycle (Table 2). In this experimental group, even if the total number of collected oocytes in this experimental group (165) wasn`t the highest comparing with two other groups (182 and 57), other parameters: Total COCs number used for IVF—84, cleavage rate—83% (*p* < 0.05), total number of expanded blastocysts—36, and development rate—21.82% (*p* < 0.05) were the best. Ovaries collected from pregnant hinds caused to be the worst for IVF comparing with two other experimental groups. Only 57 oocytes were isolated from these hinds. Moreover, other parameters were the nearest in this experimental group: Total COCs number used for IVF—48, cleavage rate—71% (*p* < 0.05), total number of expanded blastocysts—7, and development rate 12.28% (*p* < 0.05). 

#### 3.2.2. Determination of the mRNA Expression of *BCL-2, BAX, PLAC8, OCT4*, and *SOX2*

The *BAX* mRNA expression was enhanced in blastocysts obtained from the oocytes collected on the 13th day of the cycle (*p* < 0.05, Figure 2A). The *BCL-2*, *OCT4*, and *PLAC8* mRNA expression increased in blastocysts developed from the oocytes collected on the 4th day of the estrous cycle, compared to the blastocysts obtained from the oocytes collected during pregnancy (*p* < 0.01, Figure 2B–D). The mRNA expression of the *SOX2* gene was not significantly different between the blastocysts developed from the oocytes collected on the 4th day of the cycle and those developed from the oocytes collected during pregnancy (*p >* 0.05, Figure 2E). 

The nearest mRNA expression for all genes was observed in the blastocysts obtained from the oocytes collected on the 13th day of the cycle, compared to the those obtained from the oocytes collected on the 4th day of the cycle and during pregnancy (*p* < 0.01, Figure 2A,C,D), with the exception of the *BCL-2* mRNA expression, which was found similar both on the 13th day of the estrous cycle and in pregnancy (*p >* 0.05, Figure 2B), and *BAX* mRNA expression, which was the highest (*p* < 0.05; Figure 2A). 

### 3.3. Experiment 2. Comparison of the Antioxidative Potential of Red Deer Embryos during the Luteal and Follicular Stage of the Estrous Cycle and Pregnancy 

The mRNA expression of *MnSOD*, *CuSOD*, and *GPX* and the activity of SOD and GPX in blastocysts were determined. The *MnSOD, CuSOD* and *GPX* mRNA expression increased in blastocysts developed from the oocytes collected on the 4th day of the estrous cycle, compared to those developed from the oocytes collected on the 13th day and during pregnancy (*p* < 0.05; Figure 3).

The concentration of SOD (activity expressed as the percentage of inhibition rate in 15 min) and GPX (activity expressed as units per mg of protein) was the highest in blastocysts received on the 4th day of the estrous cycle. It decreased in the blastocysts developed from the oocytes collected from pregnant hinds and was the lowest in the blastocysts developed from the oocytes collected on the 13th day of the cycle (*p* < 0.05; Figure 4A,B). 

## 4. Discussion

Deer embryo production in vitro has the potential to increase valuable traits for the agricultural sector, and from a conservation perspective, it can be used as a propagation tool to improve genetic diversity in small captive populations. Our results showed that the IVF with cryopreserved semen in red deer has the highest fertilization efficiency index obtained so far. Moreover, we compared the different stages of reproductive period in hinds to select the most effective oocyte donors for IVF. The quality of embryos was evaluated by the mRNA expression of indicators, like *OCT4, SOX2, PLAC8*, and *BCL-2*, and the highest expression was observed in the expanded blastocysts developed from the oocytes collected on the 4th day of the estrous cycle. Moreover, the same embryos were characterized to show their elevated antioxidative potential. In this study, we showed the relationship between the quality and antioxidative potential of embryos. 

The quality of the blastocysts obtained from red deer was assessed by analyzing the expression of genes previously examined in cattle, sheep, and Iberian red deer [4,17]. Usually, cattle are subjected to pharmacological stimulation of follicular hormone (FSH) prior to oocyte collection for IVF [35]. So far, only one publication concerning multiple ovulation and embryo transfer in red deer has been described [36]. Nevertheless, in the case of wild or even farmed red deer, the pharmacological synchronization is associated with the stress caused by the immobilization of hinds. Based on the ultrasonographic monitoring of antral follicle development in red deer provided by [37], one, two, or three follicular waves occur during the estrous cycle in hinds. They observed that during the luteal cycle, the total numbers of follicles > 3 mm did not vary significantly by day. A single, large (> or =6 mm) follicle was usually present on all days, except for the time immediately after ovulation. However, the appearance of new follicles (> or =3 mm) was the greatest on days 1–3 and 12–14. We compared the number of oocytes and blastocysts, cleavage and developmental rates, as well as the quality of blastocysts developed from the oocytes collected on the 4th and 13th days of the estrous cycle and in pregnancy. The best-quality blastocysts were those developed from the oocytes collected on the 4th day of the estrous cycle, and it might be the effect of the earlier synchronization of hinds, especially by FSH. De Roover et al. indicated that FSH/LH stimulation in cows might have a positive effect on in vitro oocyte developmental competence, as more embryos are cultured with less amount, but presumably better-quality oocytes [35]. However, this aspect demands further study.

The antral follicle count (AFC) is used commonly as an indicator of cow fertility and according to Ireland et al. and the number of follicles positively correlated with the development rate of blastocysts [38,39]. These data are in agreement with our results because the nearest number of follicles and oocytes we received from pregnant hinds, as donor of oocytes, and further the development rate in this experimental group was the lowermost. Thus, the development rate, ranging between 12.28% in pregnant females and 21.82% in non-pregnant hinds on the 4th day of the cycle, is dependent on the reproductive status. The differences in the development rate of blastocysts have been shown previously by Dadashpour Davachi et al., who obtained higher efficiency with the ewes in the breeding season, compared with the females in anestrous phase [40]. Nevertheless, the opposite results in which the blastocyst development rates were independent of the season were presented in sika deer by the laparoscopic ovum pick-up, ranging from 22% to 34% of the total oocytes during the non-breeding and breeding seasons [41]. What is more, the highest development rate was not correlated with the cleavage rate in the same experimental group (hinds—oocytes donors form 4th day of the estrous cycle). 

The blastocysts have the ability to proliferate and differentiate into specialized cell types under controlled conditions because they are composed of embryonic stem cells [42]. The transcripts of *BAX* and *BCL-2* have been described in bovine and deer oocytes and embryos before implantation [4,19]. The higher *BAX* mRNA expression characterized embryos with worse quality and slow development [42]. The transcription factors *OCT4, NANOG*, and *SOX2* are expressed in the inner cell mass and trophoectoderm of bovine blastocysts. *SOX2* is a component of the so-called pluripotency network, in which it plays a central role along with *OCT4* and *NANOG* [43]. These genes regulate their own and each other’s expression and often bind to the same target genes [44]. The knockdown of *SOX2* led to a reduction in *NANOG* expression in bovine blastocysts, probably because of the *SOX2*-mediated regulation of *NANOG*, as mentioned above. *OCT4* is required for the maintenance of the pluripotency of the inner cell mass and is used as a marker of embryonic stem cells [44]. The expression of *OCT4* was significantly higher in the inner cell mass than in the ovine trophoblast [45]. Another developmentally important gene is placenta-specific 8 gene (*PLAC8*), which was up-regulated in hatched blastocysts, compared to the early blastocysts [46]. We received the expression of all studied factors, except for *SOX2*, increased in the expanded blastocysts developed from the oocytes collected on the 4th day of the estrous cycle. Since we obtained significant differences in the expression of factors considered as blastocyst quality indicators, we concluded that for hinds, the reproductive status of donor-females affects the quality of embryos obtained as a result of IVF. The experimental material—oocytes—were obtained from hinds at different seasons of the year, when the length of the daylight changed (between September and January). The effect of melatonin on the in vitro developmental competence of bovine oocytes has already been described by Pang et al. [47], who showed that after IVF, oocytes receiving melatonin supplementation exhibited a significantly higher blastocyst formation rate in ewes [48] and cattle [49]. 

Likewise, the protective role of melatonin on the quality and development of embryos might be attributed to its antioxidative activities, as suggested by Pang et al. [47]. There are no available results concerning the dependence between the term of oocyte collection and antioxidative potential of blastocysts. However, the incubation of poor-quality human embryos was associated with a decline in antioxidative capacity, which was higher than the antioxidative ability observed in “good” and “fair” embryos [49]. The findings suggest that the impaired embryo development might be associated with an increased generation of reactive oxygen species by the embryo [4]. Our results revealed that the mRNA expression and activity of antioxidative enzymes (SOD and GSH) increased in the blastocysts developed from the oocytes collected on the 4th day of the estrous cycle, comparing to those developed form the oocytes collected on the 13th day and during pregnancy. However, the MnSOD mRNA expression was not significantly different between the blastocysts from animals on the 4th day of the cycle and pregnant hinds, but it decreased in blastocysts from animals on the 13th day of the cycle. 

## 5. Conclusions

In summary, we showed that when the oocytes for IVF are selected depending on the stage of the reproductive period, the competence of blastocysts is changeable in hinds. The blastocyst rate was positively correlated with the mRNA expression of apoptosis markers and antioxidative potential measured by the mRNA expression and activity of SOD and GSH. The 4th day of the estrous cycle was found to be the most effective period for oocyte collection for IVF. Further studies are necessary to confirm that the reproductive status of oocyte-donor hinds determine the blastocyst competence by evaluating other oocyte and blastocysts quality markers.

## Figures and Tables

**Figure 1 animals-10-01190-f001:**
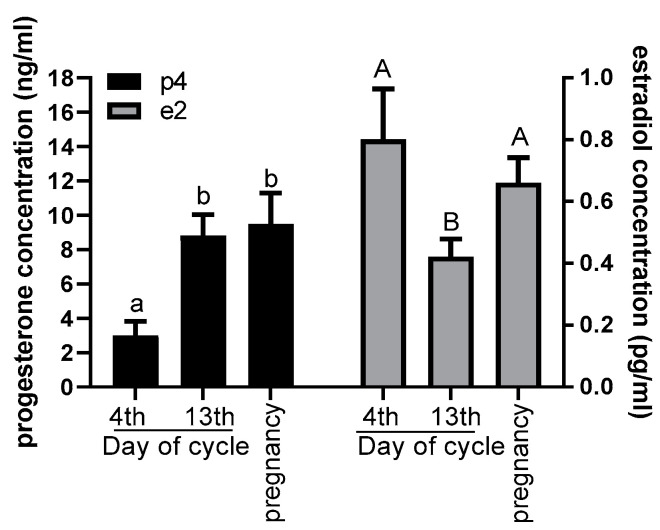
Concentrations of progesterone (left axis of graph; ng/mL) and 17-beta estradiol (right axis of graph; pg/mL) in the blood. Different letters indicate statistical differences in hormone concentrations on the 4th day versus the 13th day of the estrous cycle versus during pregnancy. *p* < 0.05 was considered significant.

**Figure 2 animals-10-01190-f002:**
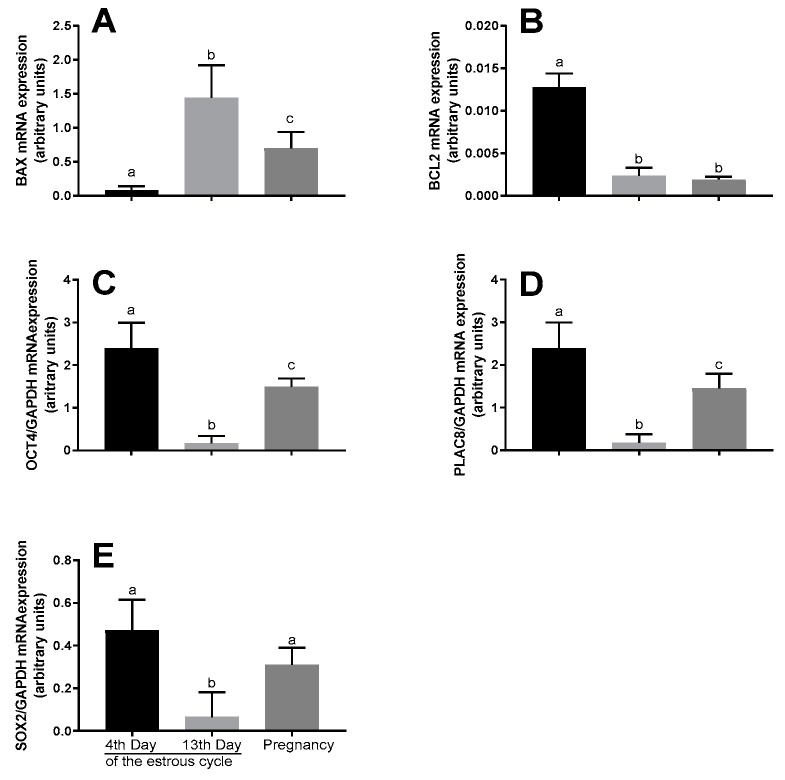
mRNA expression of *BAX* (**A**), *BCL2* (**B**), *OCT4* (**C**), *PLAC8* (**D**), and *SOX2* (**E**) in red deer blastocysts received on the 4th day versus the 13th day of the estrous cycle versus during pregnancy, as determined by real-time PCR. Data were normalized against the mRNA expression of the internal control gene glyceraldehyde-3-phosphate dehydrogenase (*GAPDH*). Bars represent the mean ± SEM. Different letters (a, b, c) indicate statistically significant differences. *p* < 0.05 was considered significant.

**Figure 3 animals-10-01190-f003:**
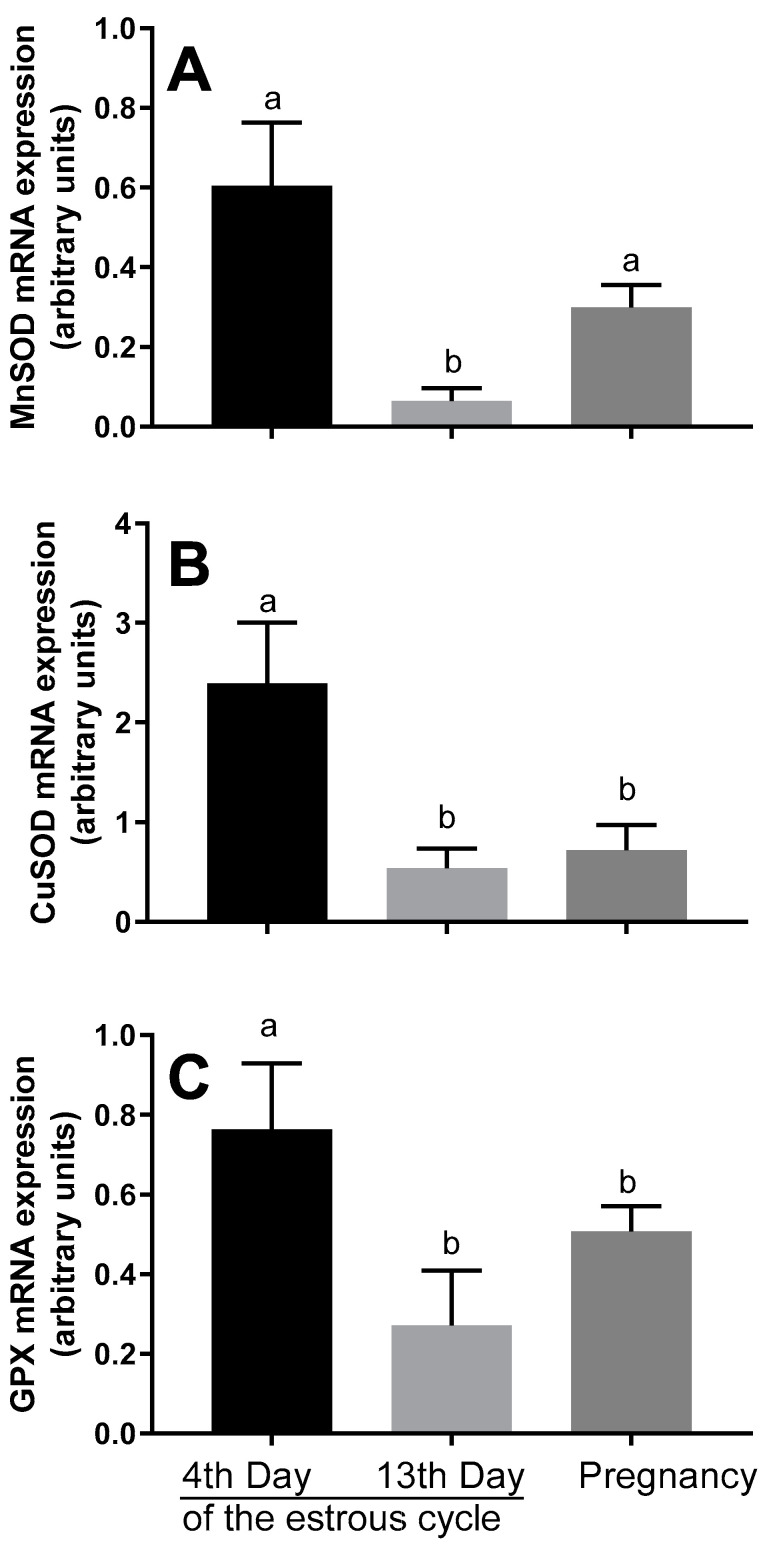
mRNA expression of *MnSOD* (**A**), *CuSOD* (**B**), and *GPX* (**C**) in red deer blastocysts developed from the oocytes collected on the 4th day versus the 13th day of the estrous cycle versus during pregnancy, as determined by real-time PCR. Data were normalized against the mRNA expression of the internal control gene *GAPDH*. Bars represent the mean ± SEM. Different letters (a, b) indicate statistically significant differences. *p* < 0.05 was considered statistically significant.

**Figure 4 animals-10-01190-f004:**
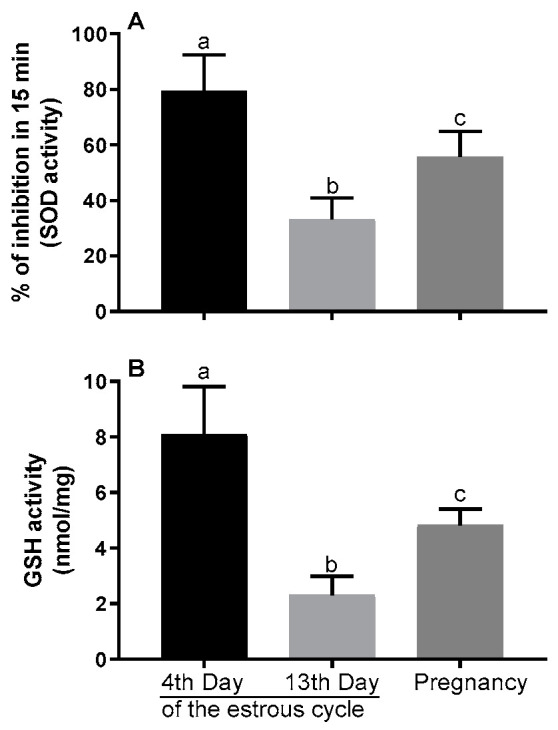
Activity of superoxide dismutase (SOD) and reduced glutathione (GSH) in red deer blastocysts developed from the oocytes collected on the 4th day versus the 13th day of the estrous cycle versus during pregnancy, as determined colorimetrically. Bars represent percentage (**A**,**B**) mean ± SEM. Different letters (a, b, c) indicate statistically significant differences. *p* < 0.05 was considered statistically significant.

**Table 1 animals-10-01190-t001:** Oligonucleotide sequences used for real-time polymerase chain reaction (PCR).

Gene	Oligonucleotide Sequences	Product Size (bp)	GeneBank
*GPX*	CCACTGGCAGGAACTTTGAT-3′ TTCCTCTTCAGGGATGGTTG-3′	137	AF080228.1
*MnSOD*	TCCTGTTCAATCGCAGTTACAGA-3′ ACGGGGTGGTGACTATCAGA-3′	162	NM_201527.2
*CuSOD*	ACACAAGGCTGTACCAGTGC-3′ TGTCACATTGCCCAGGTCTC-3′	105	NM_174615.2
*ACTB*	CCAAGGCCAACCGTGAGAAAAT-3′ CCACATTCCGTGAGGATCTTCA-3′	256	K00622
*RN18S*	AAGTCTTTGGGTTCCGGG-3′ GGACATCTAAGGGCATCACA-3′	365	AF176811
*GAPDH*	CACCCTCAAGATTGTCAGCA-3′ GGTCATAAGTCCCTCCACGA-3′	103	BC102589
*BAX*	GTGCCCGAGTTGATCAGGAC CCATGTGGGTGTCCCAAAGT	126	NM_173894.1
*BCL2*	GAGTTCGGAGGGGTCATGTG GCCTTCAGAGACAGCCAGGA	203	NM_001166486.1
*SOX2*	TGGATCGGCCAGAAGAGGAG CAGGCGAAGAATAATTTGGGGG	89	NM_001105463.2
*OCT4*	GAGAAAGACGTGGTCCGAGTG GACCCAGCAGCCTCAAAATC	101	NM_174580.2
*PLAC8*	TTTACCGCTCTGTGCCCTTT CCATGTGAACTTGACCAAGCAT	95	NM_001025325.2

**Table 2 animals-10-01190-t002:** Number of collected oocytes, cumulus-oocyte complexes (COCs), and blastocysts on particular days post in vitro fertilization and embryo development rate. Different letters indicate statistical differences in hormone concentrations on the 4th day versus the 13th day of the estrous cycle versus during pregnancy in cleavage and developmental rates. *p* < 0.05 was considered significant.

Experimental Groups—Ovaries Collected from Hinds:	Total Number of Collected Oocytes	Total COCs Number	Total Number of Blastocysts/ Development Rate	Number of Blastocysts Collected on Particular Days Post in vitro Fertilization
4th day of the cycle (*n* = 6)	165	84	36 21.82 %	Day 6th – 17 Day 7th – 7 Day 8th – 7 Day 9th – 5
13th day of the cycle (*n* = 6)	182	72	24 13.19 %	Day 6 – 7 Day 7 – 5 Day 8 – 11 Day 9 – 1
pregnant (*n* = 6)	57	48	7 12,28 %	Day 5 – 3 Day 6 –2 Day 7 – 1 Day 8 – 1

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
