# Peer review of "Antioxidative Potential of Red Deer Embryos Depends on Reproductive Stage of Hind as a Oocyte Donor"

_animals, 2020, doi:10.3390/ani10071190_

Round 1

Reviewer 1 Report

The article sent for review presents interesting research results on red deer reproduction. The manuscript requires minor additions to better understand the research methodology.
My suggestions:
Section 2.3. Semen collection, cryopreservation, and preparation for in vitro fertilization do not contain essential information necessary to replicate the experience. It is necessary to supplement the sperm movement assessment method, specify the procedure and indicate what equipment was used, whether dilution of the samples was used to evaluate the movement, at what temperature the analyzes were performed. How long did it take from collecting to freezing semen? What was the time to equilibrate the samples? What was the volume of the straws? Did the authors take into account the percentage of morphologically abnormal sperm when selecting semen (if so, this fragment should be supplemented, if no, the reason should be given)?
In subsection 2.4. In vitro fertilization is short of time and temperature for thawing semen. There is also no information on the time elapsed from thawing to IVF.
You should also correct small typos in the text, for example, L: 162, should be 'Petri'

Author Response

The authors would like to thank Editors and Reviewers for reviewing the manuscript. We used all of Editors and Reviewers suggestions in order to make our manuscript more consistent and suitable for publishing in Aanimals and submitted the responces to Reviewers in separate pdf files.

Reviewer 2 Report

Review

The manuscript submitted for review raises a rather important issue related to reproduction of the red deer, and thus the functioning of the population and its continuity. The work is an important contribution to research in the field of in vitro reproduction of deer for use in closed farming for meat production as well as for possible colonization of natural habitats for hunting. The manuscript is written fairly correctly, but contains a number of small, but significant errors, both substantive and methodological, which in my opinion should be supplemented before its publication.

Specific remarks:

Line 87-88, the authors state the phrase "most ovarian studies" - while supporting it with just one citation of the literature.

Material and methodsLine 121, 12 female deer from closed breeding were used in the study, 6 to the first experimental group and 6 to the second experimental group. The females were 3-4 years old, but no exact age should be given which should be known in farm conditions. In addition, sampling took place after pharmacological synchronization of the oestrus cycle, and the control group came from wild animals where the oestrus cycle was naturally occurring (line 139). It should definitely be stated that there are serious doubts whether the control group was correctly selected for this experiment?Another element raising quite considerable doubts is the fact of assessing the age of wild red deer females. The authors state that the ovaries were collected from 3-4 years old hinds during the hunting season, without specifying in any way the method of assessing their age and who possibly made such an assessment and what experience they had in this matter. At the same time, they say that it was a period of about 40 days of pregnancy, but they do not specify by what methods it was determined. Oestrus in female deer is quite stretched in time and based on general information it cannot be assumed that fertilization occurred in the 3rd decade of September, as juvenile hinds often have oestrus much earlier, but there is no precise age assessment to specify it accurately. In addition, the course of oestrus is determined by climatic conditions that were completely omitted in the experiment.

Line 149, The authors report the time between collection and processing for 70 minutes. It is very unlikely that after shooting, finding the animal, performing evisceration, and yet such were necessary for collecting ovaries and blood from the heart chamber and transport to the laboratory, it was all possible to do in 70 minutes. In addition, age assessment was also carried out during this period, however, as I emphasized, it is unknown by what methods.Line 168, the authors state that semen was collected from 7 wild bulls of the red deer shot in the hunting season, whose age was rated, among others based on the shape of the antler and tooth wear. The antler statement should be considered absurd, as this indicator can in no way be used to assess the age of this group of animals. While the method of age assessment based on tooth wear and characteristic changes in the registers of the premolars and molars of the jaw is practiced in hunting as well as in scientific research, there is no indication of who made such an assessment and what experience he had in this regard.Another element is the fact of collecting semen from bulls 5-7 years old, apart from the accuracy of age assessment, which raises serious doubts. According to the literature, the culmination of the individual development of bull deer falls on the age of 7-10 and in natural conditions such bulls mainly breed (as so-called herd bulls), and in my opinion research material should be collected, which would imitate the natural cycle of reproductive biology.There is definitely no assessment of the individual condition of animals used in the experiment. Individual condition, even if expressed by a simple carcass mass index, directly influences the reproductive status of animals (both males and females). What's more, the authors themselves raise this issue in the discussion, but do not pay attention to it in the research methodology, as well as the results obtained. Deer are animals that live in studs, and therefore the social hierarchy will also affect reproductive status. The authors do not state which farm animals and which wild animals were chosen for the experiment. In the case of wild animals, usually those with a lower individual condition and no offspring are intended for hunting, which is conditioned by individual and population selection in cervids, however, the authors do not indicate by which key 6 wild hinds were selected as a control.Line 387, pharmacological heat synchronization undoubtedly affects the phenomenon of stress, which can lead to multidirectional physiological changes in the body, and thus the results obtained may differ slightly from those that could be obtained in the absence of heat synchronization or possibly in natural conditions. The authors indicate this in the discussion.One should agree with the authors' suggestion that further research in this area should be carried out, taking into account a number of other methodological aspects in the selection of animals for the experiment and their comprehensive assessment of health and physiological status, which directly affects their reproductive status. Obtained results and really formulated one conclusion should be considered only for demonstrative purposes, which can be used for further research in this area.I did not assess the language and style of work, as it will probably be caught up in a possible publishing process, but in some places it raises doubts.

Final conclusion:In general, the article can be published, after adding additions to the previously described mainly methodological issues, and if this is not possible, then the methodology should be made more detailed, or I would suggest to completely omit the control group as inadequate to the results obtained from farm animals. In addition, I would suggest changing the title, because the selected experimental groups do not reflect the reproductive status of the red deer in any way, while the current wording suggests that the experiment carried out a comprehensive assessment of the reproductive status of animals.

Author Response

The authors would like to thank Editors and Reviewers for reviewing the manuscript. We used all of Editors and Reviewers suggestions in order to make our manuscript more consistent and suitable for publishing in Animals and the responces to Reviewers we uploaded as a separate files (pdf).

Round 2

Reviewer 2 Report

Certainly the corrections made clarified some methodical inaccuracies. The authors' explanations sent are substantive and do not raise any doubts. At the same time, I still have doubts about the title of this manuscript, it is difficult for me to impose a specific title on the authors, but in my opinion it is still incomplete.

I do not have any other comments and I think that the manuscript may be published.